# Sulforaphane Inhibits the Expression of Long Noncoding RNA H19 and Its Target APOBEC3G and Thereby Pancreatic Cancer Progression

**DOI:** 10.3390/cancers13040827

**Published:** 2021-02-16

**Authors:** Yiqiao Luo, Bin Yan, Li Liu, Libo Yin, Huihui Ji, Xuefeng An, Jury Gladkich, Zhimin Qi, Carolina De La Torre, Ingrid Herr

**Affiliations:** 1Department of General, Visceral & Transplant Surgery, Molecular OncoSurgery, Section Surgical Research, University of Heidelberg, 69117 Heidelberg, Germany; luo@uni-heidelberg.de (Y.L.); Bin.Yan@stud.uni-heidelberg.de (B.Y.); l.liu@uni-heidelberg.de (L.L.); yinlibo001@163.com (L.Y.); huihui.ji@uni-heidelberg.de (H.J.); Xuefeng.An@stud.uni-heidelberg.de (X.A.); gladkich@uni-heidelberg.de (J.G.); Qi@uni-heidelberg.de (Z.Q.); 2Medical Research Centre, Medical Faculty Mannheim, University of Heidelberg, 69117 Heidelberg, Germany; carolina.delatorre@medma.uni-heidelberg.de

**Keywords:** pancreatic ductal adenocarcinoma, sulforaphane, long noncoding RNAs, lncRNA-H19, APOBEC3G, TGF-β

## Abstract

**Simple Summary:**

Pancreatic ductal adenocarcinoma (PDAC) is a highly malignant tumor with poor therapeutic responses and short survival. The identification of factors that make PDAC so deadly and their targeting is important. Sulforaphane has shown promise in experimental and epidemiological studies, as well as in initial patient pilot studies. We examined the influence of sulforaphane to the largely unexplored epigenetic regulators “long noncoding RNAs” (lncRNAs). Sulforaphane modulated the expression of the lncRNAs H19, MALAT1, HOTAIR, HOTTIP and PVT1. The downregulation of the tumor promoter H19 and its target gene APOBEC3G was most significant and converged in inhibition of Smad2/TGF-β, which is involved in prevention of PDAC progression. Our data identified APOBEC3G as a new H19 target and a novel therapeutic target in PDAC, which can be inhibited by sulforaphane. The provided gene array accession numbers are important for future exploration of the suggested mechanism.

**Abstract:**

Pancreatic ductal adenocarcinoma (PDAC) is extremely malignant and the therapeutic options available usually have little impact on survival. Great hope is placed on new therapeutic targets, including long noncoding RNAs (lncRNAs), and on the development of new drugs, based on e.g., broccoli-derived sulforaphane, which meanwhile has shown promise in pilot studies in patients. We examined whether sulforaphane interferes with lncRNA signaling and analyzed five PDAC and two nonmalignant cell lines, patient tissues (*n* = 30), and online patient data (*n* = 350). RT-qPCR, Western blotting, MTT, colony formation, transwell and wound healing assays; gene array analysis; bioinformatics; in situ hybridization; immunohistochemistry and xenotransplantation were used. Sulforaphane regulated the expression of all of five examined lncRNAs, but basal expression, biological function and inhibition of H19 were of highest significance. H19 siRNA prevented colony formation, migration, invasion and Smad2 phosphorylation. We identified 103 common sulforaphane- and H19-related target genes and focused to the virus-induced tumor promoter APOBEC3G. APOBEC3G siRNA mimicked the previously observed H19 and sulforaphane effects. In vivo, sulforaphane- or H19 or APOBEC3G siRNAs led to significantly smaller tumor xenografts with reduced expression of Ki67, APOBEC3G and phospho-Smad2. Together, we identified APOBEC3G as H19 target, and both are inhibited by sulforaphane in prevention of PDAC progression.

## 1. Introduction

Pancreatic ductal adenocarcinoma (PDAC) is one of the most lethal malignancies due to its typical late diagnosis and poor prognosis [1,2,3]. Despite worldwide efforts, the treatment of patients suffering from PDAC is still unsatisfactory. Recent epidemiological data have suggested that the regular consumption of vegetables of the Brassicacea plant family, such as broccoli, Brussels sprouts, cabbage, cauliflower, kale, swede and turnip, is associated with a reduced incidence of cancer [4], including an inverse association with the risk of developing PDAC [5,6,7,8,9], as well as cancer of the breast, kidney, bladder and prostate [10,11,12,13,14,15]. The obvious anticancer activity of Brassicaceae has been linked to the high content of sulfur-containing glucosinolates [16], with a focus to glucoraphanin and its active isothiocyanate sulforaphane, which is enriched in broccoli and its sprouts [17]. These findings were underlined by positive results of patient pilot studies with sulforaphane-enriched plant extracts in non-resectable PDAC [18], and advanced prostate cancer [19]. Sulforaphane is one of the most well studied bioactive agents with antimicrobial properties against fungi, bacteria and viruses and anti-inflammatory and antioxidative activity; it also has the ability to induce detoxifying enzyme expression, cell cycle arrest, apoptosis [16], epigenetic regulation [20], and to inhibit TGF-β/Smad-induced epithelial-mesenchymal transition (EMT) [21,22,23]. Recent studies show that sulforaphane overcomes cancer stem cell features and therapy resistance in PDAC, prostate and breast cancer by attenuating the overactivation of NF-κB [24,25,26,27,28], which is mediated in part by inhibition of c-Rel by miRNA-365 [24,29]. Based on these promising data many patients have hopes to sulforaphane-rich broccoli seeds or supplements thereof, which are offered by several companies [21]. Even the development of highly active sulforaphane analogues suited for medication is in progress and were able to inhibit the progression of PDAC without side effects in our former experimental studies [30]. Therefore, the elucidation of the exact mechanisms of action of sulforaphane has great importance. 

Interestingly, Beaver et al. [31] showed that sulforaphane attenuates the expression of endogenous long noncoding RNAs (lncRNAs) in prostate cancer. lncRNAs are epigenetic regulators, whose regulation and mechanistic impact in PDAC progression is still unclear. LncRNAs are defined as noncoding RNAs longer than 200 nucleotides [32]. They are critical controllers of transcription, posttranscriptional regulation, genomic imprinting and many other biological processes [33]. Recently, high expression of the lncRNAs H19 (H19), HOTAIL, HOTTIP, MALAT1, and PVT1 in PDAC was reported [34,35,36,37,38]. Previous studies suggest that H19 is a cancer promoter, which is not only highly expressed in PDAC [39], but also in cancer of the breast [40], stomach [41], colon and rectum [42]. One putative lncRNA target may be the “virus protein apolipoprotein B mRNA editing enzyme catalytic polypeptide-like 3G” (APOBEC3G), which is a critical restriction factor for limiting retroviral infection [43]. In addition, a role of APOBEC3G in disease progression and metastasis of PDAC is described [44], as well as in cancers of the colon [45], uterus, and cervix [46]. 

In the present study, we demonstrated that sulforaphane modulates the expression of several lncRNAs in PDAC, and H19 was the most significantly downregulated lncRNA. We identified APOBEC3G as a major H19 target gene. siRNA-mediated inhibition of H19 or APOBEC3G mimicked the tumor-inhibiting effect of sulforaphane. Our data support the hypothesis that sulforaphane-mediated downregulation of APOBEC3G prevents Smad2 phosphorylation and thereby TGF-driven progression of PDAC.

## 2. Materials and Methods

### 2.1. Tumor Cell Lines

The human PDAC cell lines BxPc-3, AsPC-1, PANC-1, and MIA-PaCa2 and the human hTERT-HPNE immortalized pancreatic duct cell line CRL-4023 (Cat#CRL-4023, RRID: CVCL_C466) were purchased from ATCC (Manassas, VA, USA), BxGEM cells are described [47], and the human hepatic stellate cell line LX2 was obtained from Merck (Darmstadt, Germany, Millipore Cat# SCC064, RRID:CVCL_5792). The cells were cultured as described [48].

### 2.2. Patient Tissue

Pancreas tissue samples from 30 patients with PDAC who underwent surgical resection (including left pancreas resection, Whipple resection or total pancreatectomy) with curative intent or with brain dead who had the willing to donate organs in life time were obtained from the tissue bank of the European Pancreatic Cancer Center Heidelberg, in accordance with the regulations of the tissue bank and the approval of the Ethics Committee of the University of Heidelberg (S-708/2019). Clinical diagnoses were established by conventional clinical and histological criteria according to the World Health Organization (WHO). All surgical resections were indicated by the principles and practice of oncological therapy.

### 2.3. Reagents

DL-sulforaphane (≥95%; S4441, Sigma-Aldrich, St. Louis, MO, USA) was dissolved in DMSO to a 100 mM stock solution and stored in aliquots at −20 °C. Each aliquot was used only once immediately after thawing. The final concentration of the solvents in the media was 0.1% or less. 

### 2.4. Small Interference RNA Transfection

FlexiTube siRNAs siH19, siA3G, and AllStars Negative Control siRNA (QIAGEN, Hilden, Germany), 20 µg each, were transfected into cells with Lipofectamine 2000 (Thermo Fisher Scientific, Dreieich, Germany) according to the instructions of the manufacturer.

### 2.5. Cell Viability Assay

PDAC cell lines were resuspended at a final concentration of 10^4^/mL, and 200 µL per well of a 96-well microplate were plated. The cell viability was assessed by 3-(4,5-dimethylthiazol-2-yl)-2,5-diphenyltetrazolium bromide (MTT) assay as described previously [26].

### 2.6. Transwell Migration Assay

Transwell migration assays were performed in 24-well invasion chambers using 6.5-mm diameter inserts with 8 µm pores (Costar, Corning Incorporated, New York, NY, USA). A concentration of 50 µg/mL Matrigel^®^ matrix (Corning, Kaiserslautern, Germany) was mixed with coating buffer (0.01 M Tris pH 8.0, 0.7% NaCl) at 4 °C, and 100 µL of this solution was pipetted to each chamber. Two hours after incubation at 37 °C, the remaining liquid was removed. After transfection, 1 × 10^5^ cells were seeded into the upper chamber in serum-free culture medium, while DMEM with 20% FCS was added into the lower chambers. After 48 h of incubation, the cells were fixed with 4% paraformaldehyde, followed by staining with crystal violet staining solution. A cotton swab was used to gently wipe off the cells on the top of the membrane. Cells that invaded the lower membrane were observed microscopically at 200 × magnification. For analysis, four vision fields were randomly chosen, and the number of invaded cells was calculated by the open source computer program ImageJ (http://imagej.net/Downloads) to evaluate the invasion index.

### 2.7. Colony Forming Assay

PDAC cells were transfected with siH19 and 24 h later the cells were reseeded at a low density in 6-well plates (BxPc-3, 1000 cells/well; AsPC1, 800 cells/well, MIA-PaCa2, 400 cells/well) in triplicate, followed by incubation for 14 days without changing the cell culture medium. After fixing with 3.7% paraformaldehyde, staining with 0.05% Coomassie Blue, washing and drying overnight, the number of colonies comprising at least 50 cells was counted by microcopy as described previously [29].

### 2.8. Wound Healing Assay

Twenty-four hours after transfection, PDAC were trypsinized, seeded into 6-well plates and incubated at 37 °C until the confluence reached about 90%. The cells were serum-starved for 24 h, followed by scratching with the tip of a 10 µL pipet tip. Twenty-four and forty-eight hours later, the percentage of the gap area relative to the original area was evaluated by microscopy and ImageJ software.as described previously [29].

### 2.9. Western Blot Analyses

Western blot analysis was performed as previously described [26] by the use of the following antibodies: rabbit polyclonal antibody against APOBEC3G/A3G (Abcam, Cambridge, UK, Cat# ab54257, RRID:AB_879556) and rabbit monoclonal antibodies against GAPDH (Cell Signaling Technology, Danvers, MA, USA, Cat# 2118, RRID: AB_561053), Ki-67 (Abcam, Cambridge, UK, Cat# ab92742, RRID: AB_10562976), Smad2 (Invitrogen, Taufkirchen, Germany, Cat# 700048, RRID: AB_2532277) and p-Smad2 (Invitrogen, Taufkirchen, Germany, Cat# 400800, RRID: AB_431599). 

### 2.10. mRNA Microarray Profiling

mRNA was isolated with the RNeasy Kit (QIAGEN, Hilden, Germany) according to the manufacturer’s instructions. Microarray analyses were performed at the Microarray-Analytic Center of the Medical Faculty Mannheim using Clariom™ D Assays (Thermo Fisher Scientific, Dreieich, Germany). The accession numbers of the gene arrays are: #E-MTAB-7559 (BxPC-3, control (CO) versus sulforaphane (SF), GSE159565 (MIA-PaCa2, nonsense siRNA control (NC) versus siH19).

### 2.11. In Silico Analysis

The Gene Expression Profiling Interactive Analysis (GEPIA, RRID:SCR_018294) online database (http://gepia.cancer-pku.cn), which contains data from The Cancer Genome Atlas (TCGA, https://www.cancer.gov/about-nci/organization/ccg/research/structural-genomics/ tcga) and The Genotype-Tissue Expression (GTEx, https://gtexportal.org/home/) databases, was used to analyze the expression of APOBEC3G in tumor and normal tissue and determine its relation to patient overall survival. Venn analysis was performed with the online webtool Bioinformatics & Evolutionary Genomics (http://bioinformatics.psb.ugent.be, RRID:SCR_010251). Heatmap and volcano plots were created with the free software environment for statistical computing and graphics R Studio (https://rstudio.com/products/rstudio/) and the packages LIMMA (RRID:SCR_010943) and ggplot2 (RRID:SCR_014601). Based on the TCGA database, 177 PDAC samples including 19,590 genes were extracted using the R package “TCGAbiolinks” and were split into 2 groups of high and low expression of APOBEC3G, according to the median expression of APOBEC3G. GSEA 4.0.3 software was used for gene set enrichment analysis (GSEA). The R package GSVA was used for gene set variation analysis (GSVA) analysis in the R Studio environment. The gene set database “h.all.v7.1.symbols.gmt” (https://www.gsea-msigdb.org/gsea/msigdb) was used for enrichment analyses.

### 2.12. RNA Extraction and qRT-PCR

Total RNA was isolated with the RNeasy Mini Kit (QIAGEN, Hilden, Germany). The High-Capacity RNA-to-cDNA™ Kit (Thermo Fisher Scientific, Hilden, Germany) was used for reverse transcription. The PowerUp™ SYBR™ Green Master Mix (Thermo Fisher Scientific, Hilden, Germany) was used for RT-qPCR according to the instructions of the manufacturer. The sequences of primers are given (Appendix A). The PCR conditions were as follows: 40 cycles of denaturation: 95 °C, 15 sec; annealing: 56 °C, 15 sec; extension: 72 °C, 1 min. The fold gene expression of all PCR results were calculated using the 2^−∆∆Ct^ method in Excel (https://toptipbio.com/delta-delta-ct-pcr/).

### 2.13. Detection of H19 Expression by In Situ Hybridization

The miRCURY LNA™ microRNA Detection Kit (QIAGEN, Hilden, Germany) was used according to the instructions of the manufacturer. A H19 probe, which was labeled with digoxigenin (DIG) at the 3′ and 5′ ends, was hybridized to PDAC tissue at 54 °C for 2 h. The sequence of the H19 probe was 5′-AATGCTTGAAGGCTGCTCCGT-3′. After stringent washes, the bound H19 probe was detected with nitroblue tetrazolium/5-bromo-4-chloro-3′-indolyphosphate p-toluidine (NBT/BCIP; Vector Laboratories, Burlingame, CA, USA), which served as a substrate. For nuclear staining, the slices of patient tissue were incubated in Fast Red (Vector Laboratories, Burlingame, CA, USA) for 1 min.

### 2.14. Immunohistochemical Staining

Frozen tissue sections were stained as previously described [29,49,50]. The following antibodies were used: rabbit polyclonal antibodies against APOBEC3G/A3G, Ki-67 (both from Abcam, Cambridge, UK, smad-2, and p-Smad2 (both from Invitrogen, Taufkirchem, Germany). Goat anti-rabbit biotinylated IgG (Vector Laboratories, Burlingame, CA, USA) was used as the secondary antibody. The positive signal was measured with ImageJ (https://imagej.net/Downloads, RRID:SCR_003070).

### 2.15. Tumor Xenotransplantation on Fertilized Chicken Eggs

Fertilized eggs from genetically identical hybrid Lohman Brown chickens were obtained from a local ecological hatchery (Geflügelzucht Hockenberger, Eppingen, Germany). The transplantation of established PDAC cell lines onto the chorioallantois membrane at day nine of chick development is described [51,52]. At day 18, the embryos were gently and humanely euthanized and the volume of the tumor xenografts was measured by calipers. The volume was calculated by the use of the following formula:(1)$$V\;=\;4/3\;\times \;\pi \;\times \;r3;\;r\;=\;0.5\times \sqrt{\left(d1\times d2\times d3\right),\;}$$
d: diameter, r: radius, V: volume.

### 2.16. Statistical Analysis

The quantitative data are presented as the mean values and standard deviations from at least three independent experiments, which were performed in triplicate or multiples thereof. Differences between groups were assessed with Student’s t test and corrected for multiple comparisons with the Bonferroni-Holm method by the use of Excel (Microsoft Corporation, Redmond, WA, USA). The Pearson product-moment correlation was performed to measure the linear correlation between two variables x and y. For the gene microarray data, the JMP software provided by SAS (Heidelberg, Germany) was used. *p* < 0.05 was considered statistically significant. In the GSEA, a false discovery rate (FDR) of 25% was used to adjust for multiple testing. * *p* < 0.05; ** *p* < 0.01, *** *p* < 0.001.

## 3. Results

### 3.1. H19 Levels are Enhanced in PDAC and Inhibited by Sulforaphane

We assessed the expression of five lncRNAs, whose expression was recently detected in the PDAC cell lines BxPc-3, AsPC-1, PANC-1, MIA-PaCa2 and SW-1990 as well as in patient-derived PDAC tissue [34,35,36,37,38]. By RT-qPCR analysis we found that lncRNA H19 had very high basal expression, which was more than 2000-fold higher in BxPc-3 cells than in the two nonmalignant cell lines. Additionally, the expression of the lncRNAs HOTAIL, HOTTIP, MALAT1, and PVT1 was elevated up to 12-fold (Figure 1A). In contrast, the treatment of BxPC-3 cells for 24 h with sulforaphane, which has the chemical structure of an isothiocyanate (Appendix A), significantly decreased the mRNA levels of H19, HOTAIR, HOTTIP and PVT1, whereas the expression of MALAT1 was enhanced (Figure 1B). We would like to mention that the used sulforaphane concentration of 10 µM was determined in our former publications as suited working concentration in vitro, and a proper solvent-DMSO control has been performed [26,53]. Because the basal expression of the biological relevant cancer promoter H19 and its fold change of sulforaphane-induced downregulation were of highest relevance, we examined the mRNA levels of H19 in four other PDAC cell lines. Compared to that in BxPc-3 cells, H19 expression was lower in MIA-PaCa2, PANC-1, and AsPC-1 cells; however, H19 expression was still strong and significantly increased compared to that in control cells, as H19 expression was almost nonexistent in the two nonmalignant cell lines. Interestingly, the BxPc-3-derived gemcitabine-resistant subclone BxGEM had even higher H19 expression than the parental cells, suggesting that H19 expression increases with malignancy (Figure 1C). Notably, sulforaphane significantly decreased the expression of H19 in all of the five examined PDAC cell lines (Figure 1D, please note, the BxPc-3 experiment in Figure 1B, D is the same). To address the clinical relevance of these findings, we examined H19 expression in PDAC patient tissues (*n* = 15) by in situ hybridization and compared the expression levels to those in non-malignant pancreas tissues (*n* = 15), obtained from brain-dead organ donors. By the use of a scoring system, which is described in the figure legend, we found a significant upregulation of H19 expression in PDAC tissues compared to nonmalignant tissues (Figure 1E), as evaluated blinded by two examiners with expertise in PDAC pathology.

### 3.2. Knockdown of H19 Inhibits Tumor Progression Features

To address the effect of H19 on PDAC progression, we lipo-transfected three different H19 siRNA constructs, namely siH19_1, siH19_2 and siH19_3, along with a nonsense siRNA control in BxPc-3, AsPC-1 and MIA-PaCa2 cells. Twenty-four hours later, the expression of H19 was detected by RT-qPCR. We found that all three constructs inhibited H19 expression, but siH19_2 had the most significant and uniform repressive effect in all three cell lines (Appendix A). Therefore, we have chosen siH19_2 for all subsequent experiments and refer to it as siH19 in the following. To evaluate the effect of siH19 on viability, we lipo-transfected siH19 into BxPc-3, MIA-PaCa2 and AsPC-1 cells and performed an MTT assay. Compared to the nonsense control, H19 knockdown strongly and significantly reduced cell viability (Figure 2A). Likewise, we examined the effect of siH19 on progression features by scratch, transwell and colony forming assays and found a strong inhibition of migration (Figure 2B), invasion (Figure 2C) and colony formation (Figure 2D) in BxPc-3, MIA-PaCa2 and AsPC-1 cells.

### 3.3. APOBEC3G is Targeted by H19 and Inhibited by Sulforaphane or H19 Knockdown

By mRNA microarray and subsequent bioinformatic analysis of the array data, we identified 2633 differentially expressed mRNAs, and the top 200 candidates with *p* < 0.01 are shown in a symbolic heatmap (Figure 3A; the detailed list of differentially regulated genes can be obtained via the gene array accession number #E-MTAB-7559). To limit the results to the most significant H19-target genes, we visualized overlapping genes between our actual gene array results and our recent gene array results [30], in which gene expression in BxPc-3 cells was examined before and 24 h after sulforaphane treatment, using a Venn diagram. By this way we identified 103 common candidate genes that are regulated by siH19 and sulforaphane (Figure 3B). Next, we excluded 53 of these common candidate genes because their expression trends were inconsistent in the two examined arrays (Appendix A). Among the remaining 50 candidate genes, 25 were upregulated and 25 were downregulated, with consistent results in both array analyses. Based on a literature search in PubMed and Web of Science with the key words “cancer promoter” or “cancer suppressor”, the number of candidate genes was further restricted to 17, including five candidates that acted as tumor suppressors among the co-upregulated genes and 12 candidates that acted as tumor promoters among the co-downregulated genes (Appendix A).

To further limit the number of candidates to the most relevant target gene, we arranged the results as volcano plots by comparing the results between untreated and sulforaphane-treated BxPc-3 and nonsense siRNA- and siH19-lipofected MIA-PaCa2 cells. With this method, we identified the APOBEC3G gene, which was on the top of both volcano plots in terms of *p* value and fold change (Figure 3C). We verified this result by RT-qPCR and Western blot analysis and demonstrated that the mRNA and protein expression of APOBEC3G was significantly downregulated upon sulforaphane treatment or siH19 transfection in BxPc-3, MIA-PaCa2 and AsPC-1 cells (Figure 3D–F).

### 3.4. Knockdown of APOBEC3G Inhibits Tumor Progression Features

To obtain knowledge about APOBEC3G, we screened the TCGA and GTEx databases. We found that APOBEC3G is highly expressed in the tissues of patients suffering from PDAC but only to a minor extent in the tissues of healthy controls (Figure 4A). Most importantly, patients suffering from PDAC, who have high APOBEC3G expression, had a shorter overall survival than PDAC patients with low APOBEC3G expression (Figure 4B). Because the detailed function of APOBEC3G in cancer is still limited, we characterized its function in PDAC. For the purpose of detecting the efficiency of siRNA APOBEC3G, APOBEC3G siRNA or a control nonsense siRNA was lipo-transfected into BxPc-3, AsPC-1 and MIA-PaCa2 cells, and the expression of APOBEC3G was examined by Western blot analysis. APOBEC3G protein expression began to be inhibited 24 h after transfection, and APOBEC3G downregulation was highest at 72 h after transfection in all cell lines (Figure 5A). Likewise, the knockdown of APOBEC3G resulted in reduced cell viability, with the most pronounced effects at 72 h after siRNA-mediated inhibition of APOBEC3G (Figure 5B). To explore the migration ability, we conducted a scratch assay. After lipofection, the cells were cultured for 48 h to reach 90% confluence, followed by scratching and documentation of the closure of the gap area after an additional 24 or 48 h. We found delayed closure of the gap area in cells treated with specific APOBEC3G siRNA compared to those treated with the control nonsense siRNA construct (Figure 5C). Similarly, the number of invading cells was significantly decreased by the APOBEC3G siRNA construct 48 h after lipofection, as evident from the results of the transwell assay (Figure 5D).

### 3.5. Sulforaphane Mimics the Effects of H19 or A3G Downregulation In Vivo

For in vivo evaluation, MIA-PaCa2 cells were transplanted onto the CAM on day 9 of embryonic development. On day 14 of development, sulforaphane or a saline control was injected into CAM vessels. Likewise, MIA-PaCa2 cells were lipo-transfected in vitro with siH19, siAPOBEC3G, or a nonsense siRNA control, followed by xenotransplantation. Tumor xenografts developed and were allowed to grow until day 18 of chick development. At that time point, all embryos were humanely euthanized, followed by tumor resection, documentation of the tumor sizes and calculation of the tumor volume. A brief schematic diagram of the treatment process is shown in Appendix A. The tumor volume was significantly smaller in sulforaphane-treated eggs (18 samples) or in eggs bearing siH19 (22 samples) or siAPOBEC3G-transfected cells (21 samples) than in untreated (13 samples) or nonsense siRNA-treated control (16 samples) xenografts (Figure 6A). Immunohistochemical staining of xenograft tissue sections with an APOBEC3G-specific antibody demonstrated that sulforaphane, as well as siH19, significantly inhibited APOBEC3G expression (8 samples for each group) (Figure 6B). Additionally, immunohistochemical staining of the xenograft tissues with a human-specific antibody for the proliferation marker Ki-67 showed significantly decreased Ki-67 expression upon sulforaphane treatment or transfection of siH19 or siAPOBEC3G (10 samples for each group) (Figure 6C). We did not detect obvious side effects of sulforaphane or siH19 or siAPOBEC3G transfection, since the embryo weight was not altered significantly between the groups (Appendix A). We resected the livers from chick embryos (Appendix A) and stained the liver sections with hematoxylin and eosin (H&E). We found no evidence of necrosis, which would have been indicative of side effects (Appendix A).

### 3.6. APOBEC3G Downregulation Inhibits TGF-β-Induced Smad2 Phosphorylation

Since sulforaphane is known to inhibit TGF-β/Smad-induced EMT and thereby inhibit cancer progression [23], we analyzed whether APOBEC3G may influence TGF-β signaling. We extracted data from the TCGA database by evaluating the documented gene expression of 177 available PDAC tissues and divided them into high and low APOBEC3G expression groups according to the median expression level of APOBEC3G. By the use of the GSEA computational method, the previously obtained two groups of low and high APOBEC3G expression were compared, which led to an enrichment plot of TGF-β signaling (Figure 7A). The enrichment score (ES) was 0.52, the normalized enrichment score (NES) was 1.58, and the FDR was 0.12, which suggests that TGF-β signaling was significantly enriched in PDAC tissue with high APOBEC3G expression. These data were confirmed by performing GSVA within the R Studio environment and by setting the threshold to t > 1. TGF-β signaling was significantly upregulated in the group with high APOBEC3G expression (Figure 7B). To further highlight these findings, we analyzed whether APOBEC3G may influence the TGF-β-mediated phosphorylation of its downstream transcription factor Smad2, as described [54,55]. BxPc-3, AsPC1 and MIA-PaCa2 cells were lipo-transfected with APOBEC3G siRNA or a nonsense siRNA control. Forty-eight hours later, the expression of phosphorylated and non-phosphorylated Smad2 was examined by Western blot analysis. While all control cells expressed basal levels of phosphorylated Smad2, the expression decreased upon inhibition of APOBEC3G (Figure 7C). This inhibition was even observed under conditions of TGF-β-induced Smad-2 phosphorylation (Figure 7D). While TGF-β treatment led to increased phosphorylation of Smad2, as expected, this phosphorylation was attenuated after knockdown of APOBEC3G. Our in vivo data confirm these results because the expression of phosphorylated Smad2 was significantly downregulated in xenograft tissue derived from MIA-PaCa2 cells treated with sulforaphane, siH19 or siAPOBEC3G, as examined by immunohistochemistry and quantitative evaluation of positively stained cells (8 samples for each group) (Figure 7E). Overall, our working hypothesis suggests that sulforaphane inhibits H19 and its target gene APOBEC3G, which leads to the inhibition of TGF-β-induced Smad2 phosphorylation. However, this suggestion is still speculative and further studies are required to clearly support the suggested novel mechanism.

## 4. Discussion

In the present study, we observed that bioactive sulforaphane is able to inhibit the expression of lncRNAs, with H19 being the most significantly inhibited lncRNA. This result is in line with the described crucial role of H19 in accelerating the progression of PDAC [39,56,57,58]. We showed that H19 is highly expressed in established cell lines and patient tissue derived from PDAC. Our results are consistent with previous work indicating that H19 expression is upregulated in several tumor entities, including pancreatic cancer [56,57,59,60,61,62]. 

Our finding of sulforaphane-regulated lncRNA expression is not based on omics analysis, but just on try-and-error experiments and RT-qPCR analysis of lncRNA expression in PDAC cells before and after sulforaphane treatment. For this, we analyzed the expression of the lncRNAs H19, HOTAIL, HOTTIP, MALAT1, and PVT1, whose high expression in PDAC was recently reported [34,35,36,37,38]. By choosing H19 as strongest sulforaphane-regulated candidate lncRNA, we were able to mimic the well-known cancer inhibitory effect of sulforaphane [21] by siRNA-mediated downregulation of H19. Nevertheless, we observed that the inhibition of H19 was associated with a strongly reduced cell viability, migration, invasion and tumor growth. Our data are consistent with the notion that a reduced expression of H19 inhibited pancreatic cancer metastasis, which involved the H19-mediated regulation of the micro RNAs miR-194 and let-7 [39,56,58].

We bioinformatically selected H19 candidate target genes and have chosen APOBEC3G as the top candidate gene due to its statistically relevant high correlation and a suggested interesting biological function as a virus-induced tumor promoter [39,40,41,42]. An open question is if APOBEC3G is a direct or indirect target of H19. Additional experiments would provide information about the regulation of APOBEC3G expression by H19. However, the way by which lncRNAs regulate the transcription of target genes is quite complex. It may involve e.g., binding of H19 to histone-modifying complexes, to DNA binding proteins, including transcription factors, and even to RNA polymerase II. In other cases, it is the act of lncRNA transcription rather than the lncRNA product that appears to be regulatory [63]. Therefore, future studies may shed new light on the regulatory axis between H19 and APOBEC3G.

Next, we confirmed a function of APOBEC3G in cancer progression, because sulforaphane or H19 knockdown subsequently decreased the expression of APOBEC3G, and APOBEC3G siRNA reduced cell viability, migration, invasion and tumor xenograft growth in our PDAC models. Some other reports point to the same direction and demonstrated for example that APOBEC3G (I) inhibited anoikis, a form of programmed cell death in cells detached from extracellular matrix, by activation of Akt kinase in pancreatic cancer cells [44]; (II) was increasingly expressed by human cervical intraepithelial neoplasia and associated with disease progression [46]; (III) promoted liver metastasis in an orthotopic mouse model of colorectal cancer [52]; and (IV) was associated with resistance of mesenchymal gliomas to radiation-induced cell death [64]. In contrast, two other studies published opposite effects and linked APOBEC3G expression to favorable clinical outcomes in high-grade serous ovarian carcinoma [65] or indicated it as a tumor suppressor in cervical cancer [66]. The reason for these conflicting results is unclear, although our data support a tumor promoter function of APOBEC3G in PDAC. In this regard, we analyzed online databases and found a high APOBEC3G expression in tissues from PDAC patients but a low APOBEC3G expression in nonmalignant pancreas tissue. Most importantly, there was a strong correlation between a high APOBEC3G expression in PDAC patient tissue and a shorter overall survival. These findings, together with our bioinformatically obtained results of significantly enhanced TGF-β signaling in samples with high expression of APOBEC3G, support APOBEC3G as a novel prognostic indicator in PDAC. It is important to note, that different phenotypes determine about a pro-tumorigenic or tumor suppressive response to TFG-β [67], which may in part explain the above described discussion if APOBEC3G is a tumor promoter or rather a tumor suppressor. Regarding the cell lines used in our present study, BxPC3 is SMAD4 deficient, MIA PaCa2 is wild type, and for AsPC-1 different genotypes have been reported. These facts should be considered in further studies.

In fact, we were able to prevent TGF-β-induced Smad2 phosphorylation by siRNA-mediated downregulation of APOBEC3G in vitro. By the use of an in vivo xenograft model our results revealed that xenograft tissue after sulforaphane treatment, knockdown of H19 or APOBEC3G, exhibited decreased proliferation and expression of phosphorylated Smad2. Our data perfectly match with data from a recent report [44], which demonstrated an interplay between TGF-β receptor and APOBEC3G-induced PI3/Akt signaling [68]. Despite our preliminary results suggesting that APOBEC3G drives TGF-β signaling, these findings may still be considered as speculative. 

To confirm whether the inhibition of H19 and its target APOBEC3G would be effective in vivo, we used fertilized chicken eggs for tumor xenotransplantation, although we are aware that immunodeficient mice are the most common animal models for tumor transplantation. However, our recent studies demonstrated that pancreatic tumors rapidly xenograft in fertilized chicken eggs, while the morphology and expression patterns, progression markers and pancreatic ductal markers were comparable between primary patient tissues and their xenograft copies [51]. Also, the individual tumor environment of primary patient tumors was largely maintained in chicken egg xenografts and comparable to those obtained in mouse xenografts [53]. At the same time, however, we would like to make it clear that PDAC subcutaneous mouse or chicken egg xenograft models generally have their limitations, and do not fully reflect PDAC’s pro-fibrotic nature, immunosuppressive tumor microenvironment, and the fact that only a minority of cells within PDAC tumors are tumor cells [69]. Regarding immunodeficiency the chicken egg model resembles immunodeficient mice, because immunocompetence in birds develops only after hatching [70]. Xenografts are transplanted to the CAM, usually between days 8–9 of development, when the blood vessel network is dense enough to support the growth of a tumor xenograft. The CAM is non-innervated and allows painless tumor inoculation, growth and injections. Experiments with fertilized chicken eggs can be easily performed in any laboratory, as an animal application is not required until day 18 of embryonic development, when the xenografts have to be resected because the chick hatches on day 21. Thus, the chicken egg xenotransplantation model is well suited for short-term studies of in vivo xenograft growth for up to 10 days. By the use of this model we confirmed our in vitro data and found that siRNA-mediated inhibition of H19, APOBEC3G mimicked the sulforaphane effect in inhibition of PDAC xenograft growth. 

## 5. Conclusions

In conclusion, our data provide strong evidence for H19-mediated APOBEC3G expression, which in turn activates TGF-β/Smad2 signaling and thereby contributes to the progression of pancreatic cancer, and this process is inhibited by sulforaphane. Thus, a high expression of lncRNA H19 and APOBEC3G may be considered as novel diagnostic markers in pancreatic cancer and sulforaphane analogues may be suited for the development of new and effective drugs for treatment of PDAC and other tumor entities.

## Figures and Tables

**Figure 1 cancers-13-00827-f001:**
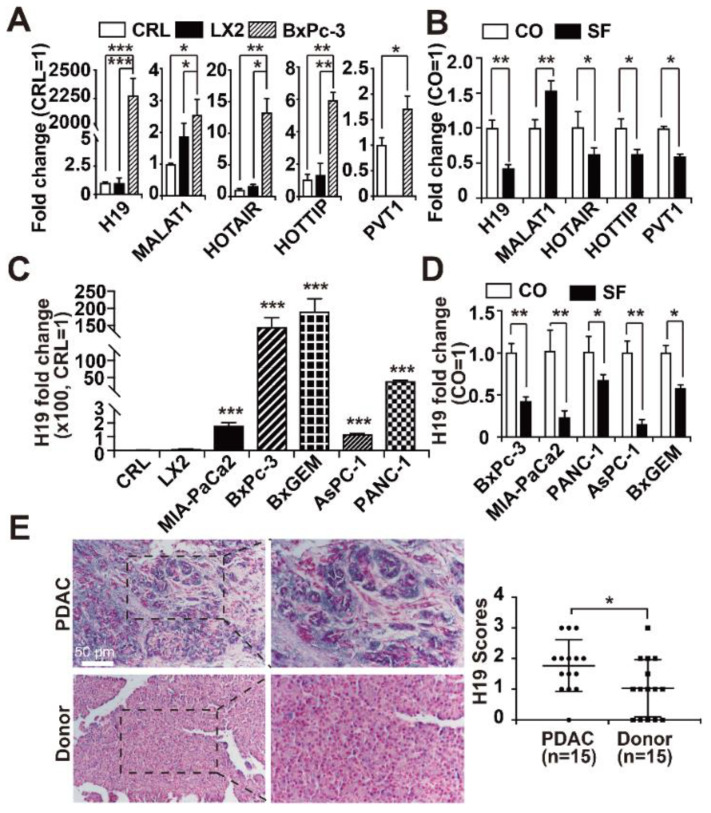
H19 expression is upregulated in PDAC and inhibited by sulforaphane. (**A**) The basal expression of the lncRNAs H19, MALAT1, HOTAIR, HOTTIP and PVT1 was detected in nonmalignant CRL-4023 (CRL) and LX2 cells and in the PDAC cell line BxPc-3 by RT-qPCR. The results are given as fold change and were normalized to the expression of CRL-4023 cells, which was set to 1. (**B**) BxPc-3 cells were left untreated (CO) or were treated with 10 µM sulforaphane (SF) for 24 h. Afterwards, the total RNA was extracted, and the expression of the lncRNAs H19, MALAT1, HOTAIR, HOTTIP and PVT1 was detected by Step One real-time qPCR. The results are expressed as fold change and were normalized to untreated control cells, whose expression was set to 1. (**C**) The expression of H19 in two human, immortalized, non-malignant cell lines CRL-4023 (CRL) and LX2, and five PDAC cell lines MIA-PaCa2, BxPc-3, BxGEM, AsPC-1 and PANC-1 cells was quantified by RT-qPCR as described above and normalized to the expression of CRL-4023 cells, because that is a non-malignant pancreatic duct cell line. (**D**) Likewise, PDAC cells as indicated were treated with 10 µM sulforaphane (SF) or were left untreated (CO), and the expression of lncRNA H19 was detected by RT-qPCR. (**E**) A DIG-labeled H19 probe was hybridized to paraffin-embedded human pancreas tissue derived from PDAC (*n* = 15) or non-malignant donated pancreas (*n* = 15), and the expression of H19 was quantified by in situ hybridization. The cell nuclei were stained with Fast Red. The dark-purple H19-positive signal was evaluated by two examiners with expertise in PDAC pathology blinded in 10 randomly chosen fields of each tissue under 400 × magnification. High, medium and low expression levels were scored as 3, 2, and 1, respectively. The absence of expression was scored with 0. * *p* < 0.05, ** *p* < 0.01, *** *p* < 0.001.

**Figure 2 cancers-13-00827-f002:**
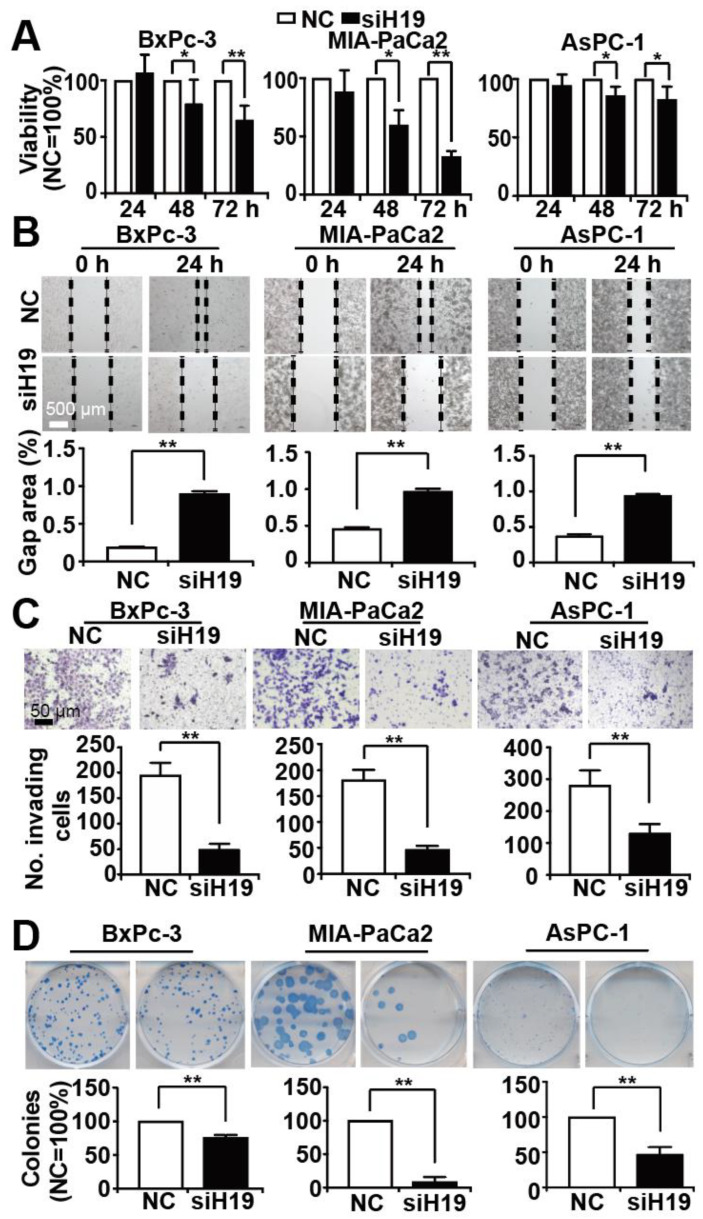
Knockdown of H19 inhibits tumor progression features. (**A**) BxPc-3, AsPC-1 and MIA-PaCa2 cells were transfected with H19 siRNA (siH19, black bars) or a nonsense siRNA control (NC, white bars), 20 µg each, using Lipofectamine 2000. Cell viability was detected by MTT assay at 24 h, 48 h and 72 h after transfection, and the data obtained with the nonsense siRNA controls of each cell line were set to 100%. (**B**) Twenty-four hours after lipo-transfection, the cells were seeded onto 6-well plates. After reaching 90% confluency 24 h later, a scratch was made to the middle of the cell layer by the use of the tip of a 10 µL pipette. The wounded region was microscopically recorded at × 100 magnification immediately after scratching (0 h) and 24 h later (24 h). The percentage of the gap area relative to the original area was analyzed by ImageJ. (**C**) Twenty-four hours after lipo-transfection, 1 × 10^5^ cells per well were seeded were seeded into the upper chamber in serum-free culture medium, while DMEM with 20% FCS was added into the lower chambers, followed by incubation for an additional 48 h. The cells were then fixed with 4% paraformaldehyde, followed by staining with crystal violet. The cells on the upper membrane of the chamber were wiped off with a cotton swab, whereas the cells on the bottom of the membrane were examined microscopically at × 100 magnification. For analysis, four vision fields were randomly chosen, and the number of invaded cells was calculated by ImageJ (http://imagej.net/Downloads, RRID:SCR_003070). (**D**) Twenty-four hours after lipo-transfection, the cells were seeded at low density (400 cells/well) into 6-well plates and cultured in regular cell culture medium for 14 days. Then, after washing and fixing in 4% PFA, cells were stained with 0.05% Coomassie blue. ImageJ was used to calculate the number of colonies consisting of at least 50 cells. * *p* < 0.05, ** *p* < 0.01.

**Figure 3 cancers-13-00827-f003:**
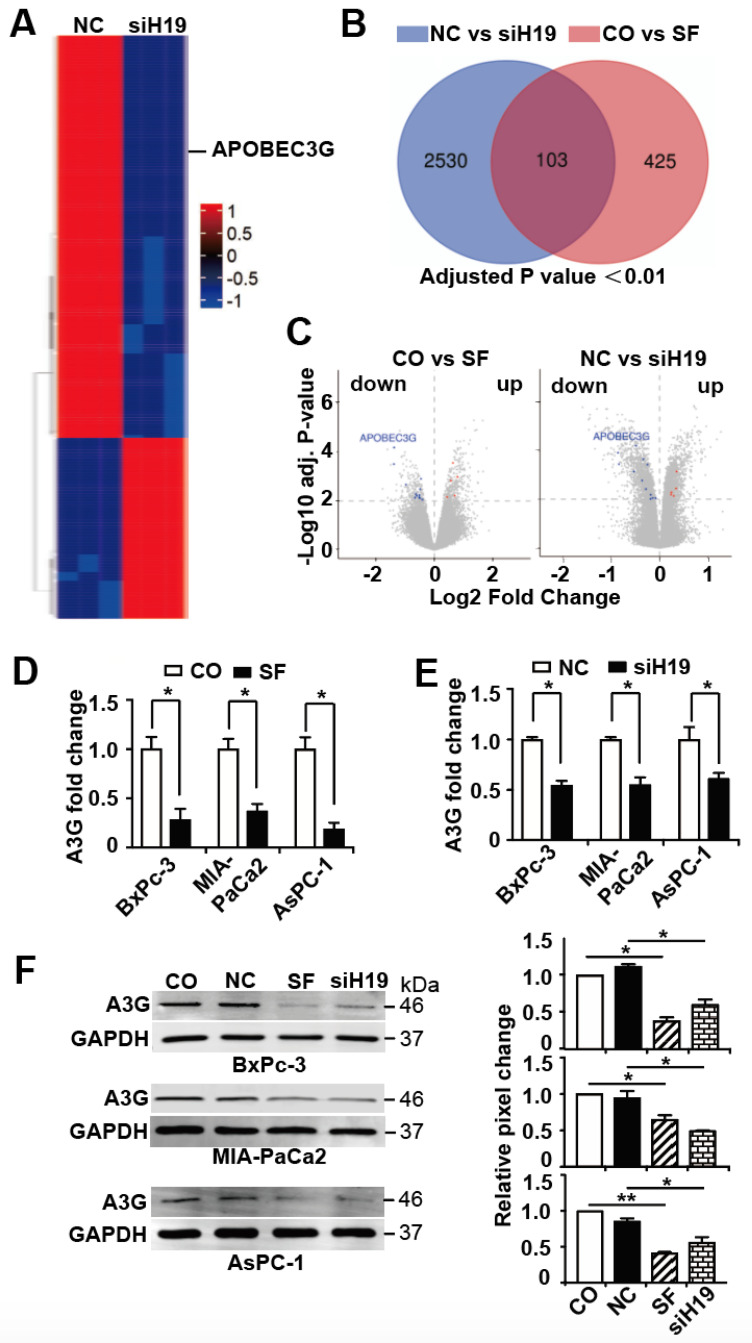
APOBEC3G is targeted by H19 and inhibited by sulforaphane or H19 knockdown. (**A**) MIA-PaCa2 cells were lipo-transfected with nonsense siRNA control (NC) or H19 siRNA (siH19), 20 µg each, and the total RNA was harvested 24 h later, followed by hybridization with a GeneChip™ miRNA 4.0 Array in triplicate and bioinformatic evaluation. The resulting heat map presents the top 200 most significantly up- or downregulated genes (red: high expression; blue: low expression). The scale from 1 to −1 indicates the relative expression. (**B**) The data from the gene array above were compared to data from our recent gene array with untreated (CO) or 24 h-sulforaphane-treated (SF, 10 µM) BxPc-3 cells and are presented as a Venn diagram. By setting the adjusted p value below 0.01, we selected 2633 candidates by comparison of siH19 and nonsense siRNA (NC)-treated MIA-PaCa2 cells and 528 candidates by comparison of sulforaphane (SF) or untreated (CO) BxPc-3 cells. Among these candidates, 103 commonly regulated gene candidates were identified. (**C**) Volcano plots were created, and the top downregulated and upregulated genes of the two comparison groups (Co versus SF and NC versus siH19) are shown. The X-axis shows the fold change of up- or downregulated candidates, while the Y-axis shows the adjusted P value. Considering both fold change and adjusted p value, APOBEC3G was the top candidate of both gene arrays. (**D**) PDAC cells were treated with 10 µM sulforaphane (SF) or were left untreated (CO), followed by harvesting the total RNA 24 h and RT-qPCR using specific APOBEC3G primers. The fold change of APOBEC3G (A3G) is shown. (**E**) PDAC cells were lipo-transfected with siH19 or a nonsense siRNA control (NC), 20 µg each, and 24 h later, the RNA was harvested, and the expression of APOBEC3G was detected by RT-qPCR. (**F**) Proteins were harvested from PDAC cells that had been lipo-transfected or treated with sulforaphane, as described above, followed by Western blot analysis and the use of a specific APOBEC3G (A3G) antibody. GAPDH served as a control for equal conditions. The protein sizes in kilodaltons (kDa) are given on the right. Statistical analysis was performed using ImageJ, and the relative pixel change was normalized to untreated control cells, as shown in the diagrams on the right. The original, crude Western blot images with molecular weight markers are shown in Appendix A. The GeneChip miRNA arrays were performed once in triplicates, and the predicted results were confirmed in three different cell lines by PCR and Western blot analysis in at least three independent experiments. The quantitative data are presented as the mean values, and standard deviations are given. * *p* < 0.05, ** *p* < 0.01.

**Figure 4 cancers-13-00827-f004:**
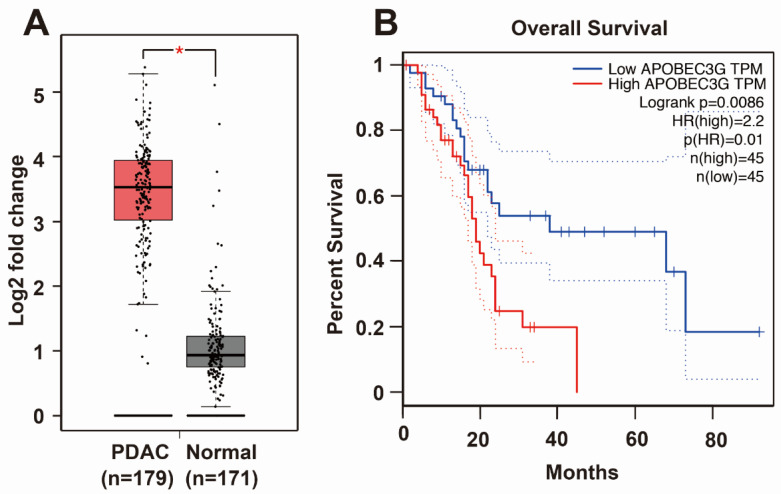
APOBEC3G is highly expressed in PDAC and related to overall survival. (**A**) TCGA and GTEx databases were utilized to evaluate the expression of APOBEC3G in human PDAC (*n* = 179) and normal pancreas (*n* = 171) tissues. (**B**) Overall survival data from patients with PDAC, with low or high APOBEC3G expression, were obtained from GEPIA, which is an online database providing patient survival analysis.

**Figure 5 cancers-13-00827-f005:**
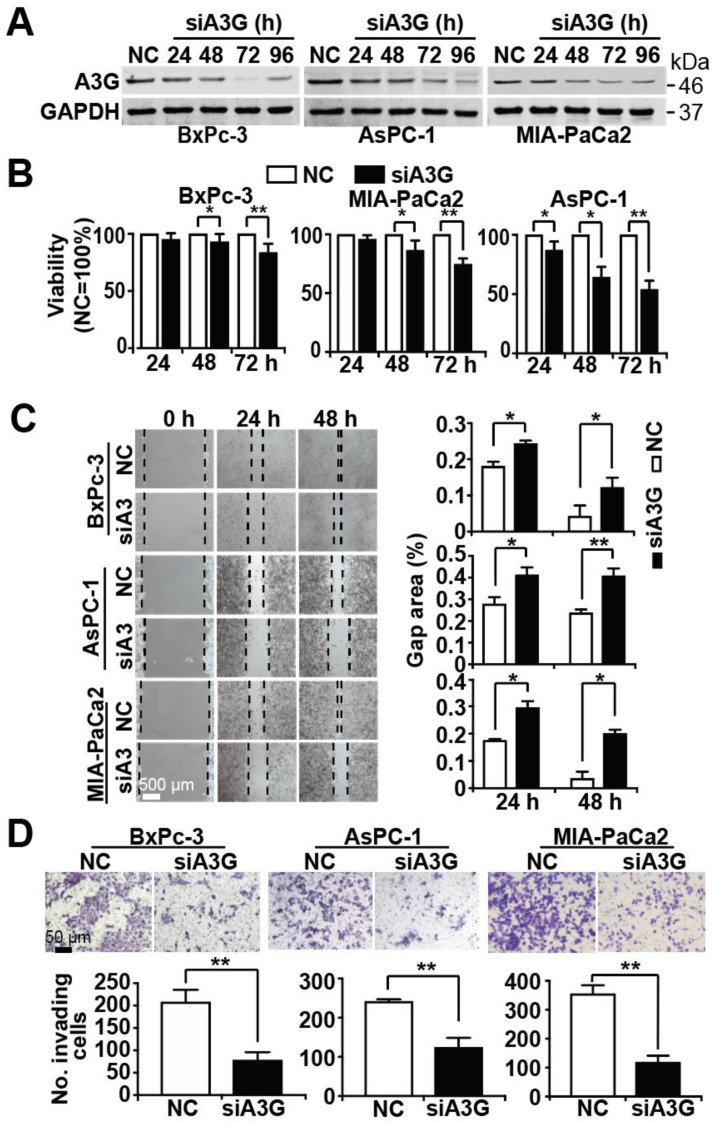
Knockdown of APOBEC3G inhibits progression features. BxPc-3, AsPC-1 and MIA-PaCa2 cell lines were lipotransfected with NC or siA3G at 20 µg each. (**A**) Proteins were harvested 24 h, 48 h, 72 h, or 96 h later, and the expression of APOBEC3G was detected by Western blot analysis. GAPDH served as a control. The protein sizes in kilodaltons (kDa) are given on the right. The original figures with all molecular weight markers are shown in Appendix A. (**B**) Cell viability was detected by the use of an MTT assay 24 h, 48 h, and 72 h after transfection, and the nonsense siRNA control data for each cell line were set to 100%. (**C**) Immediately after transfection, the cells were cultured for 24 h in 6-well plates until reaching 90% confluence. Then, a scratch was made into the middle of the cell layer with the tip of a 10 µL pipette. After an additional 24 h or 48 h, the width of the gap was examined microscopically. ImageJ was used to evaluate the percentage of the gap area relative to the original area by detecting the square of length × width. (**D**) Transfected cells at a density of 2.5 × 10^4^ were cultured in the upper chamber of Transwell plates. After 48 h, the upper cell layer was wiped off, and the cells in the bottom of the chamber were stained with 0.05% Coomassie blue. ImageJ was used to calculate the number of invading cells per field.

**Figure 6 cancers-13-00827-f006:**
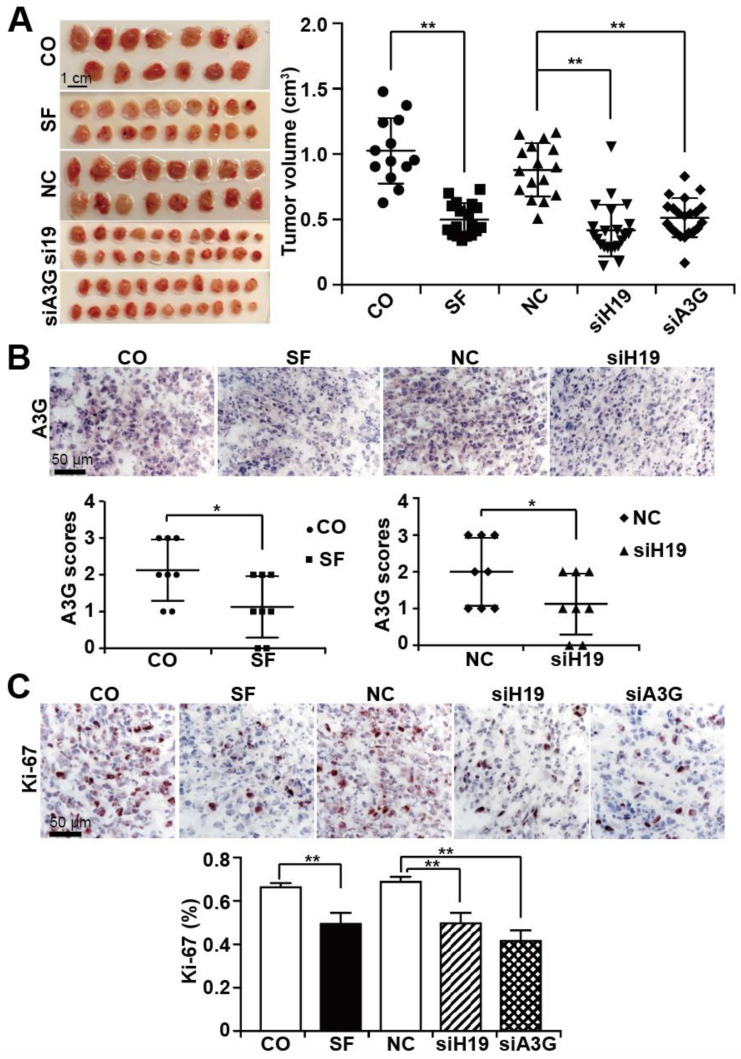
Sulforaphane mimics the effects of H19 or A3G downregulation in vivo. (**A**) MIA-PaCa2 cells were transfected with nonsense siRNA control (NC), H19 siRNA (siH19), or A3G siRNA (siA3G). Twenty-four hours later, 10^6^ cells in 50 µL Matrigel^TM^, were transplanted onto the CAM of fertilized chicken eggs on chick developmental day 9. At day 14, 100 µL of a 100 µM sulforaphane (SF) solution or 100 µL saline (CO) was injected directly into the CAM vessels that supplied the xenograft tumors. Tumor xenografts were resected on day 18 and photographed (left image). The scale bars indicate 1 cm. Please note that the images show the two-dimensional size, but the diagram the tumor volume. The individual tumor volumes and the mean volumes of each group are presented on the right. (**B**) APOBEC3G protein expression was detected by immunohistochemical staining and counterstaining of the cell nuclei with hematoxylin, followed by fluorescence microscopy at 400 × magnification. The expression levels of APOBEC3G were quantified by counting the positively stained cells in 10 randomly chosen vision fields of each tissue. High, medium, low and absent expression were scored with 3, 2, 1, and 0, respectively. (**C**) The expression of the proliferation marker Ki-67 was detected and quantified by immunohistochemistry, as described above. * *p* < 0.05, ** *p* < 0.01.

**Figure 7 cancers-13-00827-f007:**
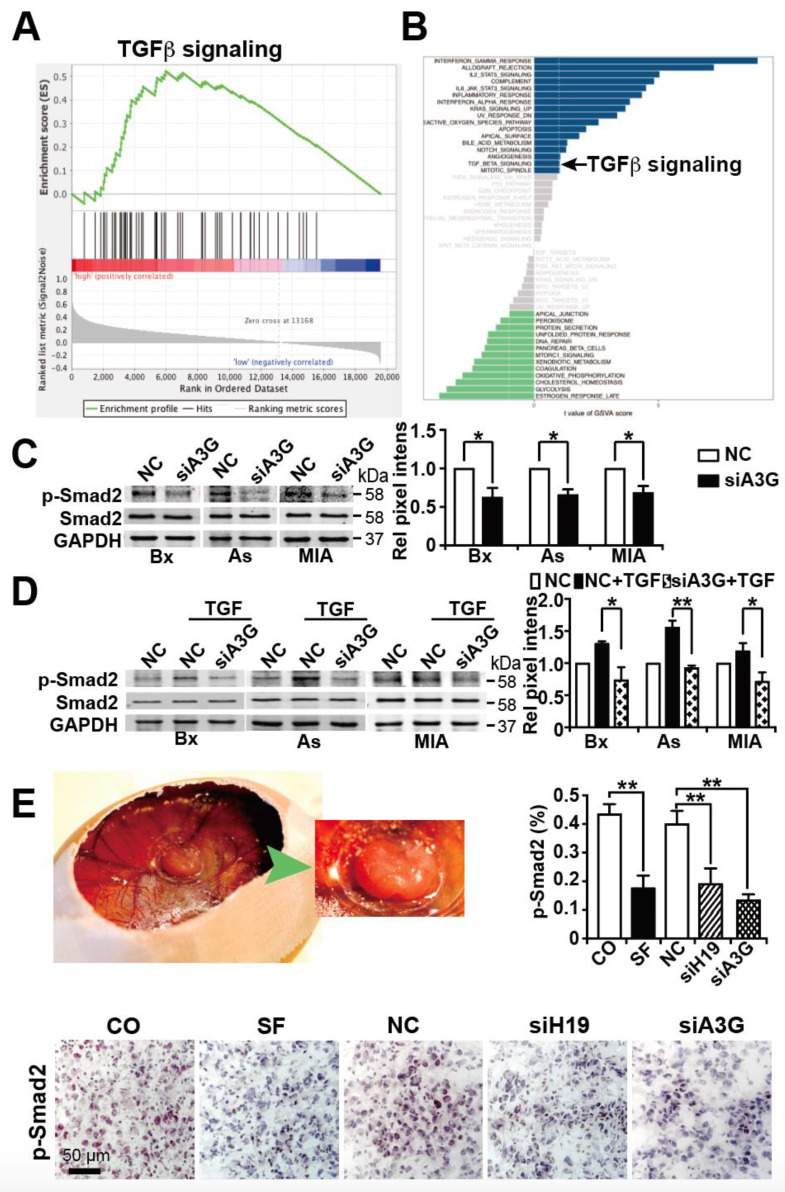
APOBEC3G downregulation inhibits TGF-β-induced Smad2 phosphorylation. (**A**) Gene expression data for 177 PDAC tissues in the TCGA database were obtained and analyzed for APOBEC3G expression; then, the samples were divided into high and low APOBEC3G expression groups. These two groups were then compared by GSEA, and the obtained enrichment plot for TGF-β signaling is shown. (**B**) In addition, GSVA within the R Studio environment was performed using the two groups obtained above and by setting the threshold to t > 1. (**C**) Forty-eight hours after lipo-transfection of APOBEC3G siRNA (siA3G) or a control nonsense siRNA control (NC) into BxPc-3, AsPC-1 or MIA-PaCa2 cells, the proteins were harvested, and the expression of phosphorylated Smad2 protein (p-Smad2) was detected by Western blot analysis. Non-phosphorylated Smad2 and GAPDH served as controls to ensure equal loading conditions. The protein sizes in kilodaltons (kDa) are given on the right. The original figures with all molecular weight markers are shown in Appendix A. (**D**) After lipo-transfection as described above, the cells were left untreated or treated with 10 ng/mL human recombinant TGF-β as indicated. Protein expression was analyzed 24 h later as described above. The original figures are shown in Appendix A. (**E**) A MIA-PaCa2 tumor xenograft growing in a fertilized chicken egg at day 18 of chick development and a magnification thereof (green arrow) is shown on the right. The expression of phosphorylated Smad2 (p-smad2) was detected by immunohistochemistry, as described in Figure 6B. * *p* < 0.05, ** *p* < 0.01.

## Data Availability

The datasets supporting the conclusions of this article are included within the article and its supplemental files and are thus available.

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
