# Peer review of "Sulforaphane Inhibits the Expression of Long Noncoding RNA H19 and Its Target APOBEC3G and Thereby Pancreatic Cancer Progression"

_cancers, 2021, doi:10.3390/cancers13040827_

Round 1

Reviewer 1 Report

The current manuscript entitled, “Sulforaphane inhibits the expression of long noncoding RNA H19 and its target APOBEC3G and thereby pancreatic cancer progression” investigated the therapeutic effects of sulforaphane in pancreatic cancer prevention. Authors have shown that H19/APOBEC3G over expressed in Pancreatic cancer, which could be potential target of sulforaphane. Though the the present study have scientific merits, however there are few points needs to be addressed.

  • Method 2.2. Authors have mentioned that they have included “30 patients with PDAC who underwent surgical resection”. Whereas in line 245 - 247  mentioned “we examined H19 expression in PDAC patient tissues (n=15) by in situ hybridization and compared the expression levels to those in healthy donor pancreatic tissues (n=15)”. Authors need to discuss the selection criteria to collect pancreatic tissue from healthy donor?
  • Did authors received consent form from heathy donor to collet pancreatic tissue? Please Provide the resection method in detail.
  • Provide the chemical structure of sulforaphane.
  • Authors need to provide brief background that why did they included sulforaphane in this study and discuss its relevance.
  • Authors have determined the mRNA expression of number of different genes by qPCR, FIGURE 1: I am wondering if CRL-4023 is parental cell for all other cells LX2, MIA-PaCa2, BxPc-3, BxGEM, AsPC-1 and PANC-1 cells ? Is they have similar phenotypes? Why did authors normalized the gene expression from other cell lines with CRL-4023 And why Did they have not used internal control house keeping gene to normalize mRNA expression of each gene in individual cell lines?
  • Figure 5: did authors performed APOBEC3G knockdown selection after transfection ? Authors need to provide the Selection procedure in method and provide Knockdown efficiency of APOBEC3G in the figure? 
  • Figure 5 A and C didn’t coincide. I am not seeing change or very meager Change after 24 and 48 hrs of transfection in the expression of siA3G, however in the figure 1c, the all cell lines adopted different Migration properties. Authors need to articulate possible mechanisms?
  • Figure 5A: quantity and normalize the WB data with loading control and plot the graph and include number of replicates.
  • Figure 6: Authors need to provide schematic diagram of tumor inoculation and treatment process.
  • Figure 5&6; Provide number of replicates generate to calculate data? 

Author Response

Reviewer #1, Cancers

The current manuscript entitled, “Sulforaphane inhibits the expression of long noncoding RNA H19 and its target APOBEC3G and thereby pancreatic cancer progression” investigated the therapeutic effects of sulforaphane in pancreatic cancer prevention. Authors have shown that H19/APOBEC3G over expressed in Pancreatic cancer, which could be potential target of sulforaphane. Though the present study have scientific merits, however there are few points needs to be addressed.

  1. Method 2.2. Authors have mentioned that they have included “30 patients with PDAC who underwent surgical resection”. Whereas in line 245 - 247 mentioned “we examined H19 expression in PDAC patient tissues (n=15) by in situ hybridization and compared the expression levels to those in healthy donor pancreatic tissues (n=15)”. Authors need to discuss the selection criteria to collect pancreatic tissue from healthy donor?

Our answer: We thank the reviewer for this important concern and apologize for the unprecise description. We would like to explain that pancreatic tissue from healthy donors means that we obtained tissue from brain dead organ donors from the tissue bank of our clinic.

Change in the manuscript: Text revision in M&Ms and Results parts, compare line 100, line 249 and line 265 of the marked manuscript.

  1. Did authors received consent form from heathy donor to collet pancreatic tissue? Please Provide the resection method in detail.

Our answer: We thank the reviewer for this important concern and apologize for the unprecise description. Healthy donor pancreatic tissue means that we obtained tissue from brain dead organ donors from the tissue bank of our clinic. The tissue obtained the tissue bank of our clinic is usually resected by left resection of the pancreas, or Whipple resection, or total pancreatectomy. The tissue is then inspected in the pathology. A piece of tissue that is not needed for pathological evaluation is either cryopreserved or embedded in paraffine. In addition, a section of the tissue block will be specially reviewed again by a pathologist to ensure that there is pancreatic cancer tissue, or normal tissue.

Change in the manuscript: Text revision in the M&Ms part, compare line 99 of the marked manuscript.

  1. Provide the chemical structure of sulforaphane.

Our answer: We thank the reviewer for this valuable hint and provided the chemical structure of sulforaphane in the Supplemental Figure S1.

Change in the manuscript: Text revision in the Results part, compare line 231 – 232 of the marked manuscript, and new supplemental Fig. S1, along with re-numbering of the old supplemental figures.

  1. Authors need to provide brief background that why did they included sulforaphane in this study and discuss its relevance.

Our answer: We thank the reviewer for this valuable suggestion and included the required information to the introduction part.

Change in the manuscript: Text change in the Introduction part, compare lines 51-70 of the marked manuscript.

  1. Authors have determined the mRNA expression of number of different genes by qPCR, FIGURE 1: I am wondering if CRL-4023 is parental cell for all other cells LX2, MIA-PaCa2, BxPc-3, BxGEM, AsPC-1 and PANC-1 cells ? Is they have similar phenotypes? Why did authors normalized the gene expression from other cell lines with CRL-4023 And why Did they have not used internal control house keeping gene to normalize mRNA expression of each gene in individual cell lines?

Our answer: We thank the reviewer for this valid question and would like to explain that CRL-4023 is a human, non-malignant, immortalized pancreatic duct cell line and LX2 is a human, non-malignant, immortalized liver stellate cell line. All other cells are human pancreatic ductal adenocarcinoma cells. Because CRL-4023 is a normal pancreatic ductal cell line, we have chosen it for normalization, to compare the different expression of lncRNA H19 in non-malignant cell lines and PDAC cell lines. We did use the house keeping gene b-actin when we performed RT-qPCR to detect the expression of H19 in different cell lines, and the fold changes of H19 expression relative to b-actin were calculated using the 2-∆∆Ct method.

Change in the manuscript: Text change in the M&M part and the legend of Fig. 1, compare line 190-191 and lines 263-268 of the marked manuscript.

  1. Figure 5: did authors performed APOBEC3G knockdown selection after transfection? Authors need to provide the Selection procedure in method and provide Knockdown efficiency of APOBEC3G in the figure?

Our answer: We agree and would like to explain that we did not perform a selection after transient liposomal transfection, but ensured the efficiency of the siRNA construct by Western blot analysis, which demonstrated significant inhibition of APOBEC3G, compare Fig. 5A.

Change in the manuscript: Text change in the Results part, compare line 364 of the marked manuscript.

  1. Figure 5 A and C didn’t coincide. I am not seeing change or very meager Change after 24 and 48 hrs of transfection in the expression of siA3G, however in the figure 1c, the all cell lines adopted different Migration properties. Authors need to articulate possible mechanisms?

Our answer: We apologize for the missing information and would like to explain that we provided the intensity ratio of each band in supplemental Fig. S5B. After transfection of siA3G, the expression of A3G was down-regulated even after 24 h and 48 h. For example, we detected the intensity ratio of each band in BxPc-3 cells, as shown in Suppl. Fig. S5B. The results were normalized to the nonsense control, which was set to 1. As shown in the figure, the intensity ratio was 0.79 at 24 h and 0.53 at 48 h after lipotransfection. These data suggest an effectice knowdown, which was high enough to justify the detected changes in migration.

Change in the manuscript: Text change in the Results part, compare line 364 of the marked manuscript.

  1. Figure 5A: quantity and normalize the WB data with loading control and plot the graph and include number of replicates.

Our answer: We agree that this information should be provided and therefore show now the original WB in the supplement Fig. S4B. All our experiments were repeated for at least three times.

Change in the manuscript: New supplemental figure S4B with accordant text revision and re-numbering of the old supplemental figures.

  1. Figure 6: Authors need to provide schematic diagram of tumor inoculation and treatment process.

Our answer: We agree and provided a brief schematic diagram in the supplemental Fig. S5.

Change in the manuscript: New supplemental figure S5 with accordant re-numbering of the old supplemental figures and corresponding text revision.

  1. Figure 5&6; Provide number of replicates generate to calculate data?

Our answer: We thank the reviewer for the suggestion and provided the required information.

Change in the manuscript: Text revision in the results part, compare lines 406–414, line 459 amd mew supplemental figure S5..

Reviewer 2 Report

The authors presented a very interesting and well written paper about the correlation between sulforaphane and the expression of some lncRNAs, that have been associated to PDAC progression. By reading all the comments about the reviewers from the previous submission, I noticed that the paper has been carefully improved, despite some suggestions were not really relevant. 

The experimental design is logic and sustains the conclusions. 

I have only some considerations and suggestions.

1) Lines 124-133: please adjust the size of the letters, they seem smaller than the rest of the text. The same at 144-146, please check that all the text has the same "size". 

2) In the discussion part, the authors decided to insert a comment by one of the previous reviewer about the correlation between transfection and knockdown efficiency; they reported the sentence "The knockdown efficiency was very high, down to about 20%, which suggests that the siRNA transfection efficiency was at least 80" (Lines 481-483), and then "However, we could not measure the transfection efficiency, because the siRNA constructs did not contain a marker sequence." (Lines 484-486). In my opinion, these sentences must be removed, because it is not necessary to test the transfection efficiency if you have a good result of a strong knockdown. Moreover, it is not correct to say that 80% of reduction means 80% of transfection efficiency, there is not a 1:1 ratio. For these reasons I strongly suggest to remove those sentences and, also, it is not necessary to write here "We would like to mention, hat we verified our data at least three times by RT-qPCR with stable results." because in the Materials and Methods section the authors have already stated that "The quantitative data are presented as the mean values and standard deviations from at least three independent experiments, which were performed in triplicate or multiples thereof." 

3) Line 566: the word "new" has been repeated twice. 

Author Response

Reviewer #2, Cancers

The authors presented a very interesting and well written paper about the correlation between sulforaphane and the expression of some lncRNAs, that have been associated to PDAC progression. By reading all the comments about the reviewers from the previous submission, I noticed that the paper has been carefully improved, despite some suggestions were not really relevant.

The experimental design is logic and sustains the conclusions. I have only some considerations and suggestions.

1) Lines 124-133: please adjust the size of the letters, they seem smaller than the rest of the text. The same at 144-146, please check that all the text has the same "size".

Our answer: We thank the reviewer for this valuable hint and revised the text accordingly.

Change in the manuscript: Text revision in the M&M part, compare lines 125-136 and 147-149.

2) In the discussion part, the authors decided to insert a comment by one of the previous reviewer about the correlation between transfection and knockdown efficiency; they reported the sentence "The knockdown efficiency was very high, down to about 20%, which suggests that the siRNA transfection efficiency was at least 80" (Lines 481-483), and then "However, we could not measure the transfection efficiency, because the siRNA constructs did not contain a marker sequence." (Lines 484-486). In my opinion, these sentences must be removed, because it is not necessary to test the transfection efficiency if you have a good result of a strong knockdown. Moreover, it is not correct to say that 80% of reduction means 80% of transfection efficiency, there is not a 1:1 ratio. For these reasons I strongly suggest to remove those sentences and, also, it is not necessary to write here "We would like to mention, hat we verified our data at least three times by RT-qPCR with stable results." because in the Materials and Methods section the authors have already stated that "The quantitative data are presented as the mean values and standard deviations from at least three independent experiments, which were performed in triplicate or multiples thereof."

Our answer: We thank the reviewer for the valuable suggestion and revised the manuscript accordingly.

Change in the manuscript: Text revision in the discussion part, compare lines 496–501 in the marked manuscript.

3) Line 566: the word "new" has been repeated twice.

Our answer: We apologize for this typing error.

Change in the manuscript: Text revision in the conclusions part, compare line 581 of the marked manuscript.

Round 2

Reviewer 1 Report

The authors have satisfied all major concerns and substantially improved the manuscript and looks self-explanatory. The manuscript can be accepted in current form.

Thanks